# Characterization of Rhizosphere Microbial Diversity and Selection of Plant-Growth-Promoting Bacteria at the Flowering and Fruiting Stages of Rapeseed

**DOI:** 10.3390/plants13020329

**Published:** 2024-01-22

**Authors:** Mengjiao Wang, Haiyan Sun, Zhimin Xu

**Affiliations:** 1School of Biological Science and Engineering, Shaanxi University of Technology, Hanzhong 723000, China; diyson2008@163.com; 2Collaborative Innovation Center for Comprehensive Development of Biological Resources in Qinling-Ba Mountains, Hanzhong 723000, China; 3Shaanxi Key Laboratory Bioresources, Hanzhong 723000, China; 4School of Nutrition and Food Sciences, Louisiana State University, Baton Rouge, LA 70809, USA; zxu@agcenter.lsu.edu

**Keywords:** *Brassica napus*, rhizosphere soil, microbial diversity, high-throughput sequencing, bacteria, fungi

## Abstract

Plant rhizosphere microorganisms play an important role in modulating plant growth and productivity. This study aimed to elucidate the diversity of rhizosphere microorganisms at the flowering and fruiting stages of rapeseed (*Brassica napus*). Microbial communities in rhizosphere soils were analyzed via high-throughput sequencing of 16S rRNA for bacteria and internal transcribed spacer (ITS) DNA regions for fungi. A total of 401 species of bacteria and 49 species of fungi in the rhizosphere soil samples were found in three different samples. The composition and diversity of rhizosphere microbial communities were significantly different at different stages of rapeseed growth. Plant-growth-promoting rhizobacteria (PGPRs) have been widely applied to improve plant growth, health, and production. Thirty-four and thirty-one PGPR strains were isolated from the rhizosphere soil samples collected at the flowering and fruiting stages of rapeseed, respectively. Different inorganic phosphorus- and silicate-solubilizing and auxin-producing capabilities were found in different strains, in addition to different heavy-metal resistances. This study deepens the understanding of the microbial diversity in the rapeseed rhizosphere and provides a microbial perspective of sustainable rapeseed cultivation.

## 1. Introduction

Rapeseed (*Brassica napus*) is one of the most important oilseed crops worldwide, which is crucial to the food industry [1]. As with most plants, the ecophysiological properties of rapeseed can be affected by changes in the rhizosphere microenvironment [2]. The rhizosphere microenvironment is a biologically active area of soil where plant roots and microorganisms interact, and it is of great importance for plant health, development, and productivity, as well as for nutrient cycling [3]. The microbiome, as the main driver of nutrient enhancement in the rhizosphere, can be beneficial or harmful to host plants [4]. Root exudates, including a range of organic acids, amino acids, sugars, and other small molecules, act as strong chemoattractants for soil microbiota [5]. Differences in the chemical composition of the exudates can occur depending on the plant species and its stage of growth [6,7]. Thus, understanding the rhizosphere’s microbial diversity is essential to understanding the plant–microbe interaction mechanism and could provide a theoretical basis for plant growth improvements and restore a good soil ecosystem [8,9]. For example, based on a study on blueberry rhizosphere microbial diversity, selected growth-promoting bacteria can significantly enhance blueberry seed germination [10].

Plant-growth-promoting rhizobacteria (PGPRs) are rhizosphere bacteria that can enhance plant growth through a wide variety of mechanisms, like phosphate solubilization and siderophore production [11]. The varieties of PGPRs have been studied for decades, and some of them have been commercialized, including the species of *Pseudomonas*, *Bacillus*, *Enterobacter*, *Klebsiella*, *Azotobacter*, *Variovorax*, *Azospirillum*, and *Serratia* [12]. As a result of the inconsistent characteristics of PGPRs, such as the ability of a PGPR to colonize the plant rhizosphere and the ability of bacterial strains to thrive under different environmental conditions, PGPR use in the agriculture industry represents only a small fraction of PGPR use worldwide [13]. The successful utilization of PGPRs is dependent on several factors, such as their survival in soil and growth-promoting properties [14,15].

In this study, the microbial diversity of rapeseed soil sample rhizospheres at different growth stages (flowering and fruiting stages) was investigated using high-throughput sequencing technology. The PGPRs in rapeseed rhizospheres were identified and isolated. The growth-promoting capabilities of these PGPR strains regarding auxin production and silicate dissolution were evaluated.

## 2. Results

### 2.1. Microbial Communities in Rapeseed Rhizosphere Soil

The microbial diversity in the rhizosphere of rapeseed at different growth stages (flowering stage, YF; silique stage, YS) was investigated using high-throughput sequencing technology. The microbial diversity in the control (YCK) soil samples was also investigated. The results were first submitted to the Sequence Read Archive (SRA). The submission ID is SUB13628190; the BioProject ID is PRJNA992224, and the BioSample Accession ID is SAMN36345610. In the three different rhizosphere soil samples, there were 401 species of bacteria and 49 species of fungi common to all of them (Figure 1). The greatest number of endemic bacteria species was 232 in the YCK soil samples, while the smallest number was 31 in the YF soil samples (Figure 1A). There were 114 endemic fungus species in the YCK soil samples, and only 44 endemic fungus species were discovered in the YF soil samples (Figure 1B).

Nine bacterial phyla (Proteobacteria, Bacteroidetes, Acidobacteria, Chloroflexi, Verrucomicrobia, Patescibacteria, Planctomycetes, Gemmatimonadetes, and Nitrospirae) with average relative abundances of above 1% were present in the YF and YS soil samples (Figure 2A,B). Proteobacteria were the most abundant phylum; the relative abundances of Proteobacteria were 24% and 30% in the YF and YS soil samples, respectively (Figure 2A,B). The relative abundances of Bacteroidetes were 19% and 17% in the YF and YS soil samples, respectively, which were higher than those of other phyla (Figure 2A,B). The relative abundance of Proteobacteria was 48% in the YCK soil samples (Figure 2C), which was significantly greater than those in the YF and YS soil samples. The relative abundance of Bacteroidetes was only 9% in the YCK soil samples (Figure 2C). The relative abundance of Rokubacteria was 5% in the YCK soil samples (Figure 2C).

Six genera (*Ellin*6067, *Sphingomonas*, *Flavisolibacter*, *Terrimonas*, UTCFX1, and *ADurb*.*Bin*063-1) had relative abundances greater than 1% in the YF and YS soil samples (Figure 2D). *Ellin6067*, *Sphingomonas*, and *Terrimonas* had relative abundances greater than 1% in the CK soil samples (Figure 2D).

Regarding fungal communities, most of the sequences were classified as Mortierellomycota (78%), Ascomycota (4%), Basidiomycota (3%), and Rozellomycota (2%) at the phylum level in YF soil samples (Figure 3A). Mortierellomycota (21%), Olpidiomycota (63%), Ascomycota (3%), and Rozellomycota (6%) were found to be the prominent fungal phyla in YS soil samples (Figure 3B). Mortierellomycota (15%), Ascomycota (31%), Basidiomycota (16%), and Glomeromycota (10%) were the four fungal phyla with an average relative abundance of above 1% in YCK soil samples (Figure 3C).

*Mortierella* was the common genera with a relative abundance greater than 1% in YF, YS, and YCK soil samples (Figure 3D). Two other genera, *Fusarium* and *Cystofilobasidium*, had a relative abundance greater than 1% (Figure 3D) in YF soil samples. *Olpidium* was another genus with a relative abundance greater than 1% in YS soil samples. There were two other gerena with relative abundances greater than 1%, *Rhizophagus* and *Ceratobasidium*, which were the prominent fungal genera in YCK soil samples as well (Figure 3D).

### 2.2. Comparison of Microbial Community Diversity in Rapeseed Rhizosphere Soil

The microbial diversity of soil samples was determined via Alpha diversity indices. Chao 1 was selected to identify community richness. The highest number of microbes was found in the YS soil samples, highlighted by the highest Chao 1 index, whereas the lowest number was found in the YCK soil samples (Table 1). There were significant differences in the Shannon indices of the fungal communities in YF, YS, and YCK soil samples (Table 1). The highest fungal Shannon index was in the YS soil samples, and the lowest was in the YCK soil samples (Table 1). There were no significant differences in the Shannon indices of bacterial communities in the YF, YS, and YCK soil samples (Table 1). The Simpson index of bacterial communities was the highest in the YS soil samples, and it was significantly different from those in the other two soil samples (Table 1). However, the Simpson index of fungus communities was the highest in the YCK soil samples, and it was significantly different from those in other soil samples (Table 1).

A Beta diversity analysis was used to evaluate differences in the YF, YS, and YCK soil samples. The species complexity at the three sampling sites was analyzed by conducting a principal coordinates analysis (PCoA) (Figure 4). In the PCoA graph, the closer the samples are, the more similar their community is. Three samples from each sampling site maintained a steady consistency (Figure 4). There was a considerable distance between YF, YS, and YCK soil samples in the graph (Figure 4). This indicates that the microbial compositions of these three soil samples are different from each other.

A PERMANOVA/Anosim suggested that the between-group differences were greater than the within-group differences (R = 0.885, *p* = 0.004 in Figure 5A; R = 0.877, *p* = 0.006 in Figure 5B). This suggests that the difference between each group was significant.

Overall, twenty-two distinct bacterial biomarkers were identified via a linear discriminant analysis (LDA) with a threshold score of ≥4.0. The YF-enriched phylotypes belonged to Chloroflexi, Verrucomicrobiota, and Bacteroidota (Chitinophagaceae) (Figure 6A). The LDA scores of YF bacterial biomarkers were between 4.216 and 5.812 (Figure 6A). Proteobacteria accounted for the majority of rhizosphere bacteria in YS (Figure 6A). The rhizosphere bacteria of YCK contained an abundance of Verrucomicrobiota and Pseudomonadota phylum, and the bacterial biomarker, which had the highest LDA score (5.091), was p_Proteobacteria (Figure 6A).

An analysis of fungal communities revealed 27 distinct biomarkers that were unevenly distributed among the microorganisms in YF, YS, and YCK rhizospheres (Figure 6B). Basidiomycota (Cystofilobasidiales) was the richest phylum in the rhizosphere mycobiota of YF, and the LDA scores of YF fungal biomarkers were between 4.305 and 4.316 (Figure 6B). The rhizosphere mycobiota of YS were rich in Rozellomycota and Olpidiomycota (Figure 6B). The LDA scores of six YF fungal biomarkers were higher than 5.0 (Figure 6B). In contrast, the YCK-specific fungi included an abundance of Basidiomycota and Mucoromycota (Glomeromycotina) (Figure 6B). The fungal biomarker that had the highest LDA score (5.236) in YCK soil samples was s_Mortierella_alpina (Figure 6B).

### 2.3. Screening the PGPR Strains in the Rhizosphere Soil of Flowering and Fruiting Rapeseed

According to the results of phosphorus and auxin production in culture by bacteria isolated from rhizosphere soil samples, 34 PGPR strains and 31 PGPR strains were isolated from the YF and YS soil samples, respectively (Appendix A). Different inorganic phosphate-solubilizing activities were found in different strains. The phosphorus concentrations in the supernatant collected from all liquid media were between 0.16 and 9.56 mg/L (Figure 7A,B). We also found that the phosphorus-solubilizing capability of 27 PGPR strains screened from YF soil samples was high (Figure 7A). The phosphorus concentrations in the supernatant collected from these 27 PGPR strains cultured in a liquid medium were higher than 3.00 mg/L (Figure 7A). The highest phosphorus concentration in the supernatant collected from YS PGPR strains cultured in a liquid medium was 2.24 mg/L (Figure 7B).

The levels of auxin produced by the strains are shown in Figure 8C,D. There were three YF PGPR strains and seven YS PGPR strains that had a high auxin production capability; the auxin concentrations in the supernatant collected from these PGPR strains cultured in a liquid medium were higher than 20.00 mg/L (Figure 7C,D).

After inoculation and incubation at 28 °C for 3 days, visible colonies with phosphorus decomposition halos were observed in the inorganic phosphorus bacteria medium inoculated with all PGPR strains. After inoculation and incubation at 28 °C for 4 days, visible colonies were observed on the silicate bacteria media inoculated with 13 PGPR strains (Appendix A). Typical phosphorus decomposition halos for PGPR strains and typical visible colonies on silicate bacteria media are shown in Figure 8A,B, respectively.

A sucrose-minimal salt low-phosphate (SLP) medium supplemented with different concentrations of heavy metals was used to test the heavy-metal resistance of the PGPR strains. Visible colonies were observed in the SLP medium after inoculation with 21 PGPR strains (Appendix A) at 28 °C for 4 days with 10 mg L^−1^ of Cu. An SLP medium supplemented with Zn in serial concentrations (100 mg L^−1^ to 700 mg L^−1^) was used to test the Zn resistance of these PGPR strains. After inoculation and incubation at 28 °C for 4 days, visible colonies were observed in the SLP medium supplemented with 700 mg L^−1^ of Zn inoculated with 17 PGPR strains (Appendix A). Pb in serial concentrations (200 mg L^−1^ to 4000 mg L^−1^) was added to SLP media to test the Pb resistance of these PGPR strains. Visible colonies of 29 PGPR strains (Appendix A) could be observed in the SLP medium supplemented with 4000 mg L^−1^ of Pb after incubation at 28 °C for 4 days. Typical visible colonies in SLP media supplemented with different concentrations of heavy metals are shown in Figure 8C–E, respectively.

All PGPR strains were inoculated and incubated at 28 °C for 4 days; visible colonies were observed on the silicate bacteria media inoculated with 13 PGPR strains (Appendix A).

According to the sequencing results, they belonged to the Pseudomonadota and Bacillota phyla. They were classified as *Klebsiella*, *Lelliottia*, *Pseudomonas*, *Stenotrophomonas*, *Bacillus*, *Fictibacillus*, or *Priestia* (Table 2).

## 3. Discussion

### 3.1. Microbial Communities in Rapeseed Rhizosphere Soil

In the rhizospheric area, rhizo-deposition is the driving force behind the initial substrate-driven community shift that exerts a great influence on rhizospheric microorganisms [16]. Plant–microbe co-evolution might lead to the active recruitment of microbiota members or at least keystone species that help functions in the plant host [17]. A variety of chemicals are secreted by different parts of the roots into the soil, acting as chemoattractants, known as root exudates [18]. They are considered the key drivers of the establishment of a host-specific microbial community in the rhizospheric zone [19]. Root exudates comprise a wide variety of compounds, such as sugars, proteins, phenols, ketones, growth hormones, flavonoids, steroids, and so on [20].

The rhizosphere soil microbial community in the flowering and fruiting stages of rapeseed was studied and was found to be significantly different from the control group. The relative abundances of seven main bacterial phyla and five main bacterial genera in the flowering and fruiting stages of rapeseed were significantly different from the control group (Appendix A). The relative abundances of Proteobacteria, Acidobacteria, Gemmatimonadetes, and other main phyla were lower in YF soil samples than in YS soil samples. Proteobacteria are well known for their role in the carbon metabolic cycle and the generation of secondary metabolites [21]. Acidobacteria are one of the most abundant soil phyla, and they previously have been shown to relate to the metabolism of organic acids in soil [22]. The phylum Gemmatimonadetes is currently recognized for its involvement in N_2_O reduction in agricultural soils [23]. Based on these results, we speculate that root secretions during the flowering and fruiting stages of rapeseed might increase the carbon metabolic cycle, organic acid metabolism, and N_2_O metabolism. This could also lead to the significantly higher relative abundance of Mortierellomycota (78%) observed in YF soil samples compared to YS soil samples [24]. There has not been enough research on root exudates in flowering and fruiting rapeseed. The difference in orthologous groups of proteins (COGs) was examined by analyzing their composition. The COG database was devised to allow for the phylogenetic classification of proteins from complete microbial genomes [25]. Significant differences in functional categories of microbial communities in metabolic pathways between samples of different groups were observed (Appendix A). The “carbohydrate transportation and metabolism” function (function J) was higher in the YF soil samples compared to the YS soil samples (Appendix A).

Rokubacteria were only found in YCK soil samples, and Glomeromycota was the only fungal phyla in YCK soil samples. Rokubacteria are found globally in diverse terrestrial ecosystems, including soils, the rhizosphere, volcanic mud, oil wells, and aquifers [26]. According to our data, the relative abundances of Rokubacteria were 0.1% and 0.2% in YF and YS soil samples, respectively. Other relevant experiments showed similar results. Rokubacteria levels have been found to be lower in soil samples after the application of rapeseed or rapeseed straw [27]. Glomeromycota is a monophyletic group of soil-borne fungi that are among the most important microorganisms on Earth, not only because they form intimate mycorrhizal associations with nearly 80% of land plants but also because they are believed to have been crucial in the initial colonization of the terrestrial realm by plants [28]. It is also notable that Glomeromycota does not colonize the roots of the Brassicaceae family [29].

The Alpha diversity index results showed that microbial richness and microbial diversity were the highest in YS soil samples compared to the other soil samples. Different plant growth stages can lead to a change in soil microbial abundance [30,31]. PCoA and iPERMANOVA/Anosim were used to assess the differences between all samples [32]. Replicates within samples were more strongly clustered in the PCoA plot than in the unweighted plot (Figure 4). An R-value closer to 1 in PERMANOVA/Anosim means that the difference between groups is significant and is greater than the difference within groups [33]. The *p*-value was less than 0.05, which means the reliability of the test was good. The results suggest that the difference between groups was significant. A LefSe was used to determine the magnitude of species and variation in each group by identifying biomarkers that differ significantly in abundance between the two groups [34]. An LDA score of >4 was considered significantly different. It was found that the core rhizobiome in the soil collected at different rapeseed growth stages was significantly different (Figure 6).

The community composition, Alpha diversity, Beta diversity, and other characteristics of rhizosphere microorganisms at different stages showed a significant difference. The COG distribution of YCK soil samples was compared with YF soil samples. It was found that YF soil samples were highly enriched in carbohydrate transportation and metabolism, cell wall/membrane/envelope biogenesis, general function prediction only, and defense mechanisms (Appendix A). However, RNA processing and modification, chromatin structure and dynamics, amino acid transportation and metabolism, lipid transportation and metabolism, cell motility, signal transduction mechanisms, intracellular trafficking/secretion/vesicular transportation, and the cytoskeleton were lower than those in the control group (Appendix A). The YS soil samples had a stronger defense mechanism compared to the control group, while amino acid transportation, metabolism, and cell motility were lower than those in the control group (Appendix A). There are two types of metabolism in the annual cycle of plants, namely, nitrogen metabolism and carbon metabolism [35]. In the early stages of vegetative growth, nitrogen metabolism is mainly consumptive metabolism [36]. During this period, nitrogen absorption and assimilation remain active. Organic nutrients are consumed more but accumulated less, thereby increasing the need for fertilizer and water, especially nitrogen [37]. From flower bud formation to fruit ripening, nutrient accumulation is greater than nutrient consumption. At this stage, carbon is mainly stored [38]. By interacting with the rhizosphere microenvironment, the metabolic processes of plants at different growth stages can affect the community structure and diversity of rhizosphere microorganisms [39,40].

### 3.2. Screening the PGPR Strains in the Rhizosphere Soil of Flowering and Fruiting Rapeseed

Plant-growth-promoting rhizobacteria (PGPRs) have multiple beneficial mechanisms for plant growth promotion. PGRPs act as a source of metabolites, enzymes, nutrient mobilization, biological pesticides, disease resistance, bioremediation, and heavy-metal detoxification [41].

In this experiment, a total of 65 PGPR strains were screened. The auxin production capability of YS31 (*Klebsiella* sp.) was the strongest among the groups. The genus *Klebsiella* was considered a PGPR that could promote plant growth by increasing its tolerance to salinity [42]. The auxin concentration in the supernatant collected from YS31 strains cultured in a liquid medium was 101.51 mg/L. Of the 13 PGPR strains, those that can dissolve potassium belong to *Pseudomonas frederiksbergensis*, *Pseudomonas* sp., *Fictibacillus* sp., *Lelliottia amnigena*, and *Klebsiella* sp. genera. *Pseudomonas* and *Fictibacillus* are important PGPRs with phosphorus- and silicate-solubilizing capacities and auxin production functions [43,44,45]. YS23 strains belong to *Lelliottia amnigena* and can dissolve potassium, but their auxin-producing and phosphorus-dissolving abilities are not significant. *Lelliottia amnigena* strains are generally associated with plants, food, and environmental sources [46,47]. This is the first time that *Lelliottia aminigena* has been reported as a PGPR.

Heavy metals alter soil properties, which can directly or indirectly influence agricultural systems. Heavy-metal toxicity in soil constitutes a substantial hazard to all living beings in the environment [48,49]. Thus, PGPR-assisted bioremediation is a promising, eco-friendly, and sustainable method for eradicating heavy metals [50]. PGPR strains with heavy-metal resistance belonged to 12 taxa (*Bacillus* sp., *Bacillus subtilis*, *Priestia aryabhattai*, *Priestia megaterium*, *Pseudomonas flavescens*, *Pseudomonas reinekei*, *Pseudomonas* sp., *Pseudomonas frederiksbergensis*, *Lelliottia amnigena*, *Klebsiella* sp. *Enterobacter* sp., and *Stenotrophomonas maltophilia*). *Pseudomonas frederiksbergensis* and *Lelliottia amnigena* are commonly recognized as plant growth promoters with heavy-metal resistance [51,52,53,54]. This is the first study to report *Pseudomonas frederiksbergensis* and *Lelliottia amnigena* as PGPRs with heavy-metal resistance. According to the existing results and other similar research, we can speculate that the practical application of PGPR strains could promote rapeseed growth, increase the amount of nutrients available and absorbed by rapeseed in the soil, and help rapeseed plants better adapt to a heavy-metal environment.

## 4. Materials and Methods

### 4.1. Collection of Soil Samples

Twenty rhizosphere soil samples were collected from rapeseed roots in the flowering stage (CV. Fangyou 777) on 10 March 2023, as described in a previous study [55]. The soil samples were collected from 5 to 20 cm below the soil surface surrounding the rapeseed stems. The soil samples were mixed and named YF. The YF soil samples were divided equally into three parts, named YFN1, YFN2, and YFN3. Twenty rhizosphere soil samples were collected from rapeseed roots in the fruiting stage (CV. Fangyou 777) on 15 May 2023 in the same sampling area. The soil samples were collected from 5 to 20 cm below the soil surface surrounding the rapeseed stems. The soil samples were mixed and named YS. The YS soil samples were divided equally into three parts, named YSN1, YSN2, and YSN3. The geographical location of the sampling sites is shown in Table 3. The control (YCK) soil samples were also collected at 5 to 20 cm under the ground surface without any growing vegetation beside the rapeseed planting area on 15 May 2023. They were also mixed and divided equally into three parts homogeneously. These three YCK soil samples were named YCK1, YCK2, and YCK3. The soil’s pH value was determined using a pH meter (PHS-3C, INESA Scientific Instrument Co., Ltd., Shanghai, China) from a soil solution with a soil sample/distilled water ratio of 1:5. The pH values of YFN1, YFN2, and YFN3 soil samples were determined, and the mean and standard derivation of these pH values was expressed as the YF pH value. The pH value of YS was determined using the same method.

### 4.2. Analysis of DNA Sequences of Microbes in Soil Sample

The genome DNA of microbes in the soil samples was extracted using a DNA extraction kit (Fast DNA Spin Kit for Soil, MP Biomedicals, Santa Ana, CA, USA). The 16S rDNA and ITS genes were amplified by specific primers (515F/806R and ITS3-F/ITS4R) (Invitrogen, Carlsbad, CA, USA, respectively). The PCR reaction solution consisted of 25 μL of 2× Premix Taq (Takara Biotech, Dalian, China), 1 μL of each primer (10 mM), 3 μL of DNA (20 ng/μL) template, and 21 μL of double-distilled H_2_O. The reaction was conducted in a BioRad S1000 instrument (Bio-Rad Laboratory, Hercules, CA, USA) with a thermos-cycling program of 5 min at 94 °C for initialization, 30 cycles of 30 s denaturation at 94 °C, 30 s annealing at 52 °C, 30 s extension at 72 °C, and 10 min of final elongation at 72 °C. DNA libraries were set up using the NEBNext Ultra DNA Library Prep Kit for Illumina (New England Biolabs, Ipswich, MA, USA). After the evaluation of library quality, the final sequences were output by an Illumina HiSeq 2500 platform (Illumina, Inc., San Diego, CA, USA).

Sequences with high similarity (>97%) were assigned to the same operational taxonomic unit (OTU) to represent a species. The chimera sequences and singleton OTUs were removed at the same time, and then statistical analyses of paired-end raw reads, paired-end clean reads, raw tags, clean tags, length of clean tags, GC percent of clean tags, and OTU number were carried out. For each representative sequence, the Silva database (https://www.arb-silva.de/, accessed on 11 June 2023) was used to annotate taxonomic information. Then, an OTU taxonomy synthesis information table was obtained for the final analysis. These Hiseq sequencing results containing double-ended sequence data (pairwise. Fastq fles) were submitted to the Sequence Read Archive (https://submit.ncbi.nlm.nih.gov/subs/sra/, accessed on 23 June 2023), and an accession number was obtained.

### 4.3. Screening and Isolation of PGPR Strains in Different Stages of Rapeseed Growth

A soil suspension was prepared to isolate PGPRs in a dilution cascade. The YF soil sample (5 g) was added to a sterilized flask and mixed with 50 mL of sterilized distilled water. Subsequently, serial dilutions were prepared up to a concentration of 10^–4^. An aliquot of 100 μL of each dilution was placed in a beef extract–peptone medium. After incubation at 28 °C for 18 h, the monoclonal microbes formed in the medium were further purified on a Petri dish using the streak method. The purified monoclonal microbes were named YF1-n. Using the same method, other purified monoclonal microbes were screened from YS soil samples and named YS1-n.

For PGPR strain screening, three inoculation points from each purified monoclonal microbes were used for cultivation on inorganic phosphorus medium (HB8670; Hope BioTech, Jinan, Shandong, China) Petri dishes at an incubation temperature of 28 °C for 7 days. The experiment was repeated in triplicate for each strain. Strains that had a soluble phosphorus circle were selected as PGPR strains for the follow-up experiments.

The phosphorus- and silicate-solubilizing and auxin-production capabilities of PGPR strains were determined using the methods of Wang et al. [10]. The PGPR strains were inoculated into a liquid inorganic phosphorus bacteria medium (HB8670-1; Hope BioTech, Jinan, Shandong, China) at 28 °C for 3 days. The incubated medium was centrifuged at 4 °C at 6000× *g* for 15 min. The supernatant was collected to determine the soluble phosphorus content using the molybdenum blue method [56]. To evaluate the silicate dissolution of the PGPR strains, three points from each PGPR strain were inoculated into silicate bacteria agar medium (HB8548; Solarbio, Beijing, China). After incubation at 28 °C for 4 days, the colony diameter was recorded to determine the silicate-solubilizing capability. The PGPR strains were inoculated into liquid beef extract–peptone medium with 100 mg/L L-Tryptophan (T0011; Solarbio, Beijing, China) and incubated at 28 °C for 3 days. The incubated medium was centrifuged at 4 °C at 6000× *g* for 15 min. The amount of auxin in the supernatant was determined using the Salkowski colorimetric method to evaluate auxin-producing capabilities [57,58].

The medium for the testing heavy-metal tolerance of PGPR strains was the SLP medium (sucrose, 1%; (NH_4_)_2_SO_4_, 0.1%; K_2_HPO_4_, 0.05%; MgSO_4_, 0.05%; NaCl, 0.01%; yeast extract, 0.05%; pH 7.2) supplemented with 10 mg/L of Cu as CuSO_4_·5H_2_O [57]. SLP medium supplemented with Zn (as ZnSO_4_) in serial concentrations (100 mg/L to 700 mg/L) and SLP medium supplemented with Pb (as Pb(NO_3_)_2_) in serial concentrations (200 mg/L to 4000 mg/L) were used to test Zn and Pb tolerance of PGPR strains, respectively. The PGPRs were inoculated into an SLP medium with the lowest concentration of heavy metal at 28 °C for 4 days. If visible colonies were observed, then the PGPRs were inoculated into the SLP medium with a higher concentration of heavy metals at 28 °C for 4 days until there were no visible colonies in the medium.

All PGPR strains were incubated in liquid beef extract–peptone medium at 28 °C for 24 h, then the supernatant was removed. The cell pellets were collected for DNA extraction. DNA extraction and 16S rDNA amplification were carried out via the same method described by Wang et al. [8]. The DNA in each PGPR was extracted according to the manufacturer’s instructions (TIANamp Bacteria DNA Kit, DP302; Tiangen BioTech, Beijing, China). The 16S rDNA genes were amplified by 27F (5′-AGA GTT TGA TCC TGG CTC AG-3′) and 1492R (5′-GGT TAC CTT GTT ACG ACT T-3′) primers (Invitrogen, Carlsbad, CA, USA). The PCR reaction solution consisted of 25 µL of 2 × Premix Taq (Takara Biotech, Dalian, China), 2 µL of each primer (10 mM), 1 µL of DNA (20 ng/mL) template, and 22 µL of double-distilled H_2_O. The solution was amplified using a BioRad S1000 (Bio-Rad Laboratory, Hercules, CA, USA) with a thermocycling program of 5 min at 94 °C for initialization, 40 cycles of 30 s for denaturation at 94 °C, 30 s annealing at 56.2 °C, 90 s extension at 72 °C, and final elongation for 10 min at 72 °C.

The 16S rDNA sequence data were compared with corresponding sequences in the GenBank database (www.blast.ncbi.nlm.nih.gov/Blast.cgi/, accessed on 23 June 2023), and the PGPR strains were classified. Based on the general classification information, the types of PGPR strains were searched, and the sequences of type strains were obtained from EZ BioCloud (https://www.ezbiocloud.net/, accessed on 23 June 2023). After comparison with the 16S rDNA sequence data, the representative sequence of PGPR strains in each species was selected and submitted to the NCBI (www.submit.ncbi.nlm.nih.gov/subs/, accessed on 23 June 2023) website to obtain the GenBank accession number [15].

### 4.4. Data Analysis

All the experiments were performed in triplicate. The Alpha diversity was determined by analyzing the complexity of the diversity using three indices, including Chao1, Shannon, and Simpson indices. These three indices were calculated using QIIME2 (https://qiime2.org/, accessed on 22 June 2023) and expressed as the mean and standard error. The relative abundance of the microbial community was also expressed as the mean and standard error.

Bray–Curtis and weighted and unweighted UniFrac Beta diversity indices were calculated using QIIME2 (https://qiime2.org/, accessed on 22 June 2023) and displayed in R software (V2.15.3). A PCoA was performed to obtain the principal coordinates and visualize complex, multidimensional data. A previously obtained distance matrix of weighted or unweighted UniFrac between samples was transformed into a new set of orthogonal axes, in which the maximum variation factor was demonstrated by the first principal coordinate, the second maximum variation factor by the second principal coordinate, and so on. The PCoA was visualized by the qiime2 and ggplot2 packages in R software (V3.2.0).

The “vegan” package in R was used for PERMANOVA/Anosim. PERMANOVA/Anosim was used following the Liang’s method [33]. Significant differences between different species were determined via a linear discriminant analysis (LDA) effect size (LEfSe) (https://github.com/biobakery/lefse/, accessed on 22 June 2023) with two as the default filter value for the LDA score. COG functional prediction was carried out following the method of Galperin et al. [34].

The phosphorus-solubilizing and auxin-producing capabilities were expressed as means and standard error.

## 5. Conclusions

The microbial diversity of the rhizosphere in a rapeseed field during flowering and fruiting stages was analyzed in this study. The community composition, Alpha diversity, Beta diversity, and other characteristics of rhizosphere microorganisms exhibited significant differences at different stages. Our results suggested that the rhizosphere’s microbial community structure is closely related to the plant development stage. Sixty-five PGPR strains were isolated and identified. PGPR strains differed in their capacity to solubilize inorganic phosphorus in media and to produce auxin. Thirteen PGPR strains could solubilize inorganic silicate, and some of them also exhibited heavy-metal tolerance. This study can improve the understanding of the rhizosphere microbiome of rapeseed during flowering and fruiting stages and can be very useful for research on PGPR strains. In the future, plant secretions and their effect on the rhizosphere’s microbial community structure will be investigated, and the growth-promoting effects of PGPR strains on rapeseed will be studied.

## Figures and Tables

**Figure 1 plants-13-00329-f001:**
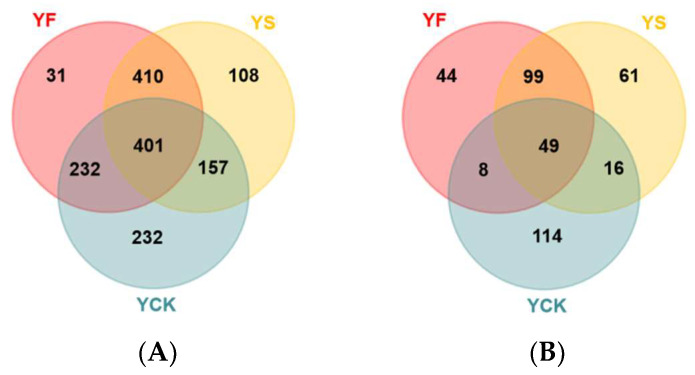
Venn diagrams of common and endemic species in rhizospheres from YF, YS, and YCK samples. (**A**,**B**) Venn diagrams of common and endemic bacterial and fungal species, respectively.

**Figure 2 plants-13-00329-f002:**
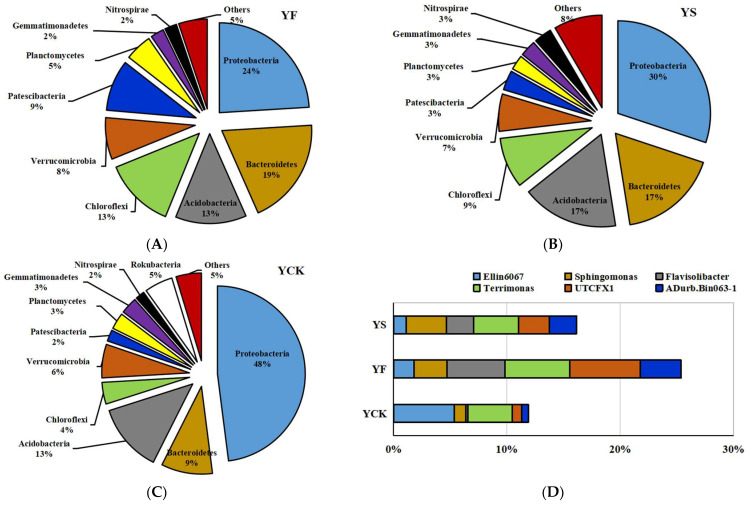
The distribution of bacteria with a relative abundance greater than or equal to 1% in YF, YS, and YCK soil samples. Panels (**A**–**C**): bacterial phyla from YF, YS, and YCK, respectively. (**D**) Bacterial genera found in the three soil samples.

**Figure 3 plants-13-00329-f003:**
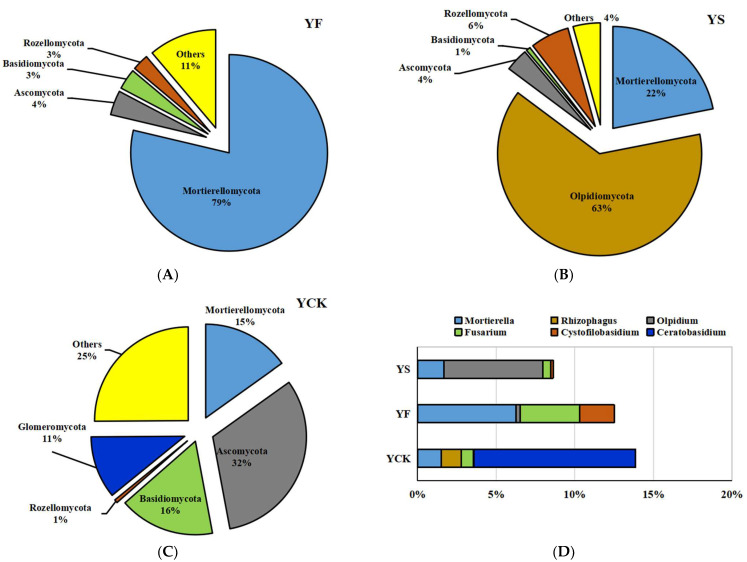
The distribution of fungi with a relative abundance greater than or equal to 1% in YF, YS, and YCK soil samples. Panels (**A**–**C**): fungal phyla from YF, YS, and YCK, respectively. (**D**) Fungal genera found in the three soil samples.

**Figure 4 plants-13-00329-f004:**
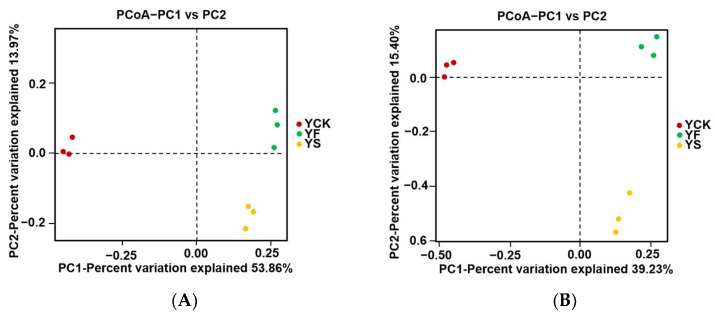
Differences in microorganism community structures in YF, YS, and YCK soil samples. Panels (**A**,**B**) show the differences in bacterial and fungal community structures, respectively.

**Figure 5 plants-13-00329-f005:**
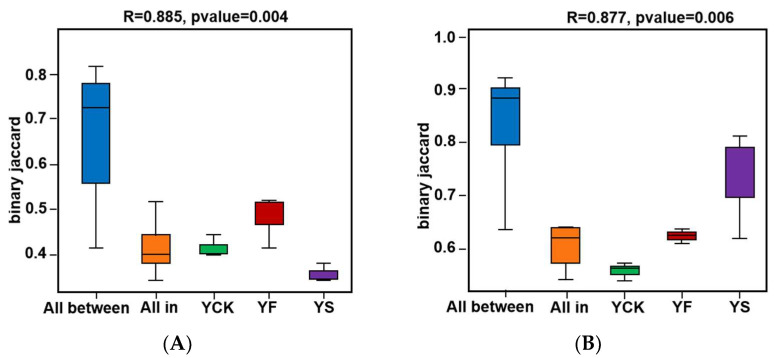
PERMANOVA/Anosim of the microbial community structure in YF, YS, and YCK soil samples based on the Bray–Curtis distance matrix. (**A**) PERMANOVA/Anosim of bacterial community structures. (**B**) PERMANOVA/Anosim of fungal community structures. The Beta distance data of samples between all groups are shown in the box diagram above “All between”. The Beta distance data of samples within all groups are shown in the box diagram above “All within”.

**Figure 6 plants-13-00329-f006:**
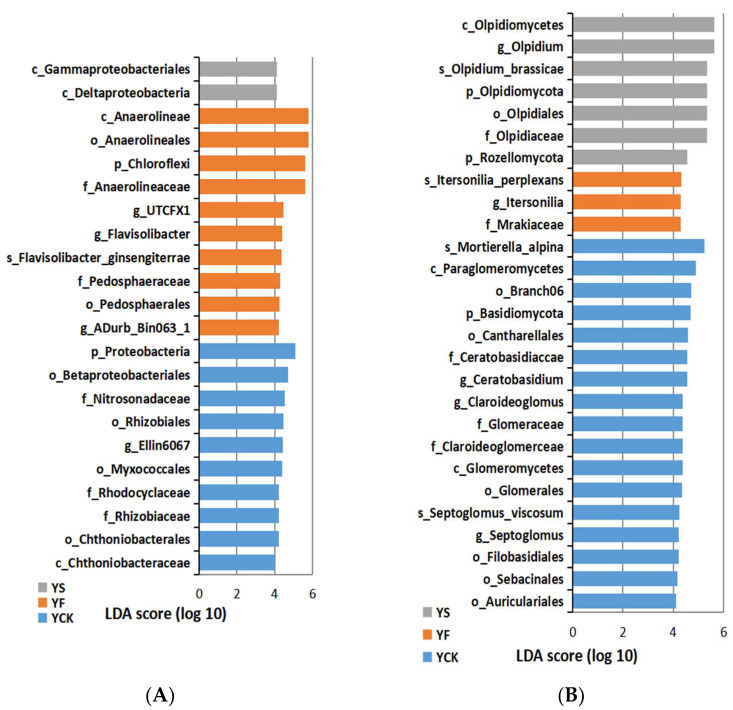
LEfSe (linear discriminant analysis effect size) of differentially abundant phylum (p), classes (c), orders (o), families (f), genera (g), and species (s) of microorganisms in the rhizosphere at different stages of rapeseed growth. (**A**) is the result of bacteria LEfSe of differentially abundant classes, and (**B**) is the result of fungi LEfSe of differentially abundant classes. Note: The LDA (linear discriminant analysis) threshold score in the figure is equal to or greater than 4.0.

**Figure 7 plants-13-00329-f007:**
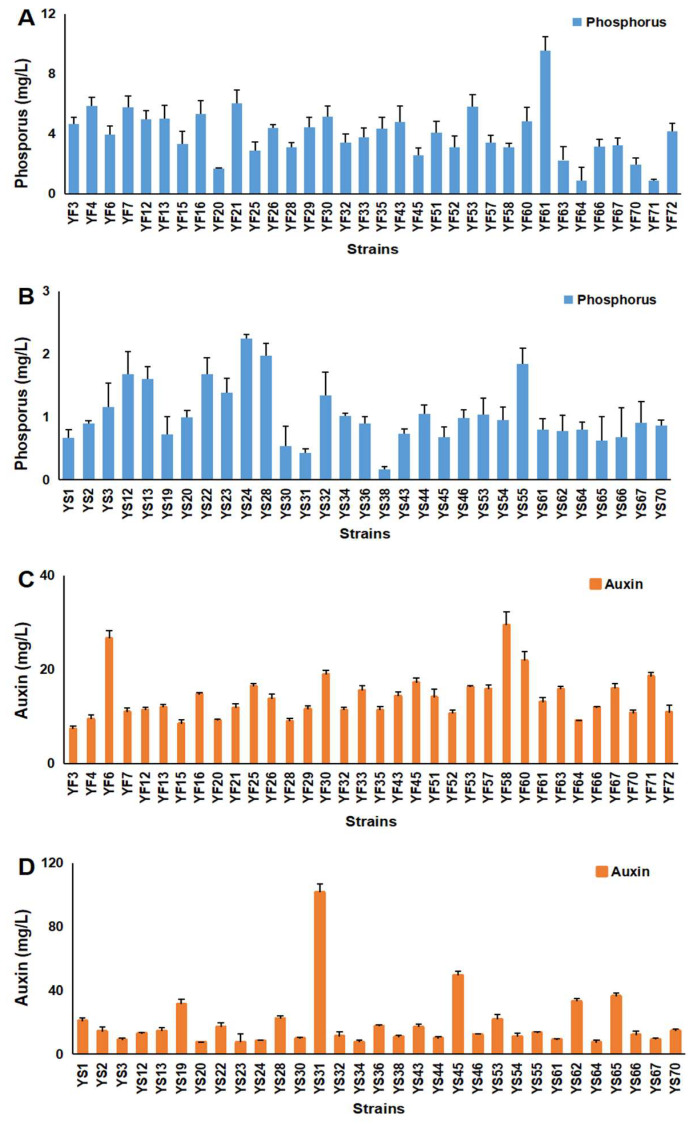
Phosphorus and auxin production capability analysis of isolated PGPR strains. (**A**) Phosphorus dissolution of YF PGPR strains. (**B**) Phosphorus dissolution of YS PGPR strains. (**C**) Auxin dissolution of YF PGPR strains. (**D**) Auxin dissolution of YS PGPR strains. Note: Vertical bars indicate the standard errors of data.

**Figure 8 plants-13-00329-f008:**
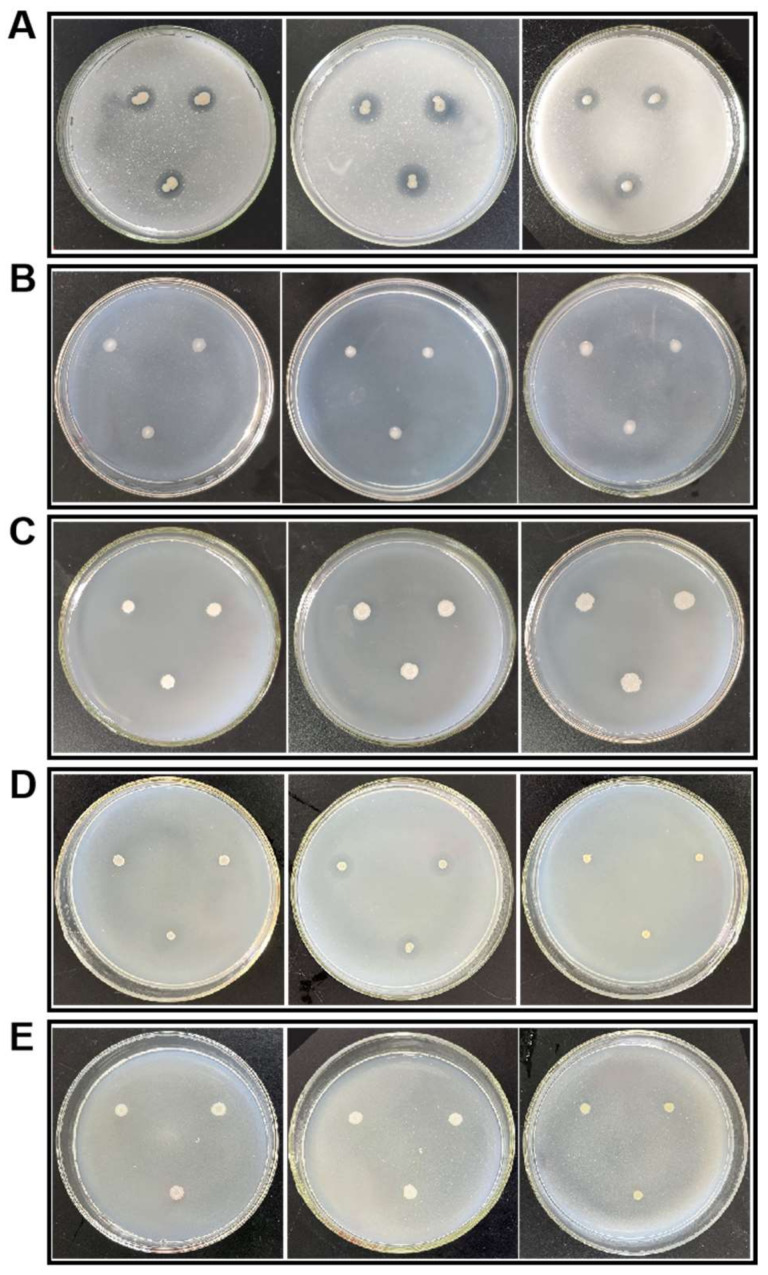
Phosphorus- and silicate-solubilizing capabilities and heavy-metal resistance analysis of isolated PGPR strains. Visible colonies on the inorganic phosphorus bacteria medium and silicate bacteria medium are shown in Figure (**A**,**B**), respectively. Visible colonies on an SLP medium supplemented with 10 mg L^−1^ of Cu, 700 mg L^−1^ of Zn, and 4000 mg L^−1^ of Pb are shown in Figure (**C**–**E**), respectively.

**Table 1 plants-13-00329-t001:** Alpha diversity index of microorganisms at the sampling sites *.

Sample	Bacteria		
	Chao1	Shannon	Simpson
YCK	708.000 ± 10.186 a	0.991 ± 0.003 a	7.995 ± 0.200 a
YF	821.376 ± 54.649 b	0.989 ± 0.001 a	7.946 ± 0.194 a
YS	1036.071 ± 31.969 c	0.995 ± 0.000 a	8.637 ± 0.053 b
	Fungus		
	Chao1	Shannon	Simpson
YCK	127.259 ± 13.351 a	0.875 ± 0.077 b	4.323 ± 0.641 a
YF	144.446 ± 11.752 a	0.510 ± 0.087 c	2.711 ± 0.223 b
YS	178.531 ± 8.342 ab	0.899 ± 0.086 a	2.387 ± 0.111 b

* Each determination was carried out in triplicate. The diversity index is indicated by the mean and standard deviation. a, b, and c mark which groups are significantly different (*p* < 0.05).

**Table 2 plants-13-00329-t002:** Classification of identified plant-growth-promoting rhizobacteria in different soil samples.

Scheme	Accession Number	Sequence Length (bp)	Related Type Strain	Type Strain Name	NCBI Taxonomy ID	Similarity to Type Strain (%)
YS22	OR234767	1112	*Acinetobacter* sp.	-	-	-
YS36	OR234768	1127	*Enterobacter* sp.	-	-	-
YS31	OR234769	949	*Klebsiella* sp.	-	-	-
YS23	OR234770	1163	*Lelliottia amnigena*	DSM 4486T	61,646	99%
YF28	OR234771	845	*Pseudomonas azotoformans*	DSM 18862	47,878	99%
YS12	OR234772	1207	*Pseudomonas flavescens*	DSM 12071	29,435	99%
YF13	OR234773	1185	*Pseudomonas frederiksbergensis*	DSM 13022	104,087	99%
YS66	OR234774	1042	*Pseudomonas jessenii*	DSM 17150	77,298	99%
YF66	OR234775	1118	*Pseudomonas reinekei*	DSM 18361	395,598	99%
YS24	OR234776	1150	*Pseudomonas umsongensis*	DSM 16611	198,618	99%
YF15	OR234777	1086	*Pseudomonas* sp.	-	-	-
YS20	OR234778	1196	*Stenotrophomonas maltophilia*	DSM 50170	40,324	99%
YS67	OR234779	1191	*Stenotrophomonas* sp.	-	-	-
YS13	OR234782	1020	*Bacillus subtilis*	ATCC 6051	1423	99%
YS44	OR234783	1149	*Bacillus* sp.	-	-	-
YF3	OR234784	1040	*Fictibacillus* sp.	-	-	-
YS65	OR234785	1056	*Priestia aryabhattai*	DSM 21047		99%
YF30	OR234780	1200	*Priestia huizhouensis*	KCTC 33172	1,501,239	99%
YF57	OR234781	1142	*Priestia megaterium*	ATCC 14581	1404	99%

**Table 3 plants-13-00329-t003:** Information of sampling sites and pH of samples.

Sample	Longitude	Latitude	Altitude (m)	pH
YCK	106°39′58″ E–106°39′40″ E	32°57′29″ N–32°57′43″ N	618.5 m–619.2 m	6.56 ± 0.087
YF/YS	106°39′7″ E–106°39′21″ E	32°57′56″ N–32°56′41″ N	619.5 m–620.5 m	6.50 ± 0.012

## Data Availability

Data is contained within the article and Appendix A.

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
