# Peer review of "Characterization of Rhizosphere Microbial Diversity and Selection of Plant-Growth-Promoting Bacteria at the Flowering and Fruiting Stages of Rapeseed"

_plants, 2024, doi:10.3390/plants13020329_

Round 1

Reviewer 1 Report (New Reviewer)

Comments and Suggestions for Authors

The content and the results of this article are interesting. The article can be split into two parts: i. the study of the microbiome (bacteria and fungi) associated to the rhizosphere of rapeseed plants in two diverese phenological phases. ii the selection and characterization of Plant Growth Promoting Rhizoacteria (PGPR) isolated from the rhizosphere  of rapeseed plants. In my opinion the e study should be published in two distinct article as there is no reason to merge them as the selection of PGPR does not depend on the study of microbioma. As a matter of fact the Authors separated the Discussion section into two distinct subsections.

Graphically' results are presented clearly but the English style of the text is poor and it  was hard to  interpret some sentences that have to be rephrased. Technical terms are often inappropriate. 

There is a funfamental methodological aspect that nust be addressed. It is not clear what is the Control in the experiment; which kind of soil and in which sense it can be regarded as a Control?

I suggested to abbreviate the Title 

For more detailed text editings, comments and suggestions see notes in the text (Attached PDF file)

Detailed Comments

  1. The original article  has to be split into two distinct articles :
  • Characterizations of rhizosphere microbioma of rapeseed plants
  •  Selection of candidate Plant Groth Promoting Rhizobacteria from the rhizosphere microbioma of rapeseed 

B. In the first of the two putative  articles the Authors have to make  clear what they considered as a Control . They  compared the rhizosphere microbiomaa of rapseeed at two different  phenological stages. What was the control and what was its scope?           

C.  In preparing the manuscripts of these two putative articles I suggest the Authors to ask a colleague of mother tongue
for help. I tried to suggest some changes  (see notes in the text, attached file available also for the Authors).

Comments on the Quality of English Language

English language is quite poor . It was very hard to interpret what the Authors wanted to say. Some sentences have to be rephrased

Author Response

Reviewer 1

Comments and Suggestions for Authors

The content and the results of this article are interesting. The article can be split into two parts: i. the study of the microbiome (bacteria and fungi) associated to the rhizosphere of rapeseed plants in two diverese phenological phases. ii the selection and characterization of Plant Growth Promoting Rhizoacteria (PGPR) isolated from the rhizosphere  of rapeseed plants. In my opinion the e study should be published in two distinct article as there is no reason to merge them as the selection of PGPR does not depend on the study of microbioma. As a matter of fact the Authors separated the Discussion section into two distinct subsections.

-Thank you for providing us an opportunity to revise the manuscript. We have revised the manuscript as suggested. The revised parts are highlighted using blue fonts.

Characterization the diversity of rhizosphere microorganisms in different growth stages of rapeseed is an very way for us to understand the rhizosphere microenvironment of rapeseed, to further obtain soil microbial resources, and to develop useful microbial strains for practical applications.

Putting these two results together in one article is not contradictory.

After understanding the background of microbial diversity in soil, it is more convincing to develop growth promoting microorganisms

This research approach makes theoretical research more closely integrated with practical research.

The “Results” part in the manuscript was divided into three parts according to the data analysis approach. There were: 2.1. Microbial communities in rapeseed rhizosphere soil, 2.2. Comparison of microbial community diversity in rapeseed rhizosphere soil,  2.3. Screening the PGPR strains in rhizosphere soil of flowering and fruiting stages fields of rapeseed.

In the “Discussion” part, there were two part: 3.1. Microbial communities in rapeseed rhizosphere soil, 3.2. Screening the PGPR strains in rhizosphere soil of flowering and fruiting stages fields of rapeseed.

This arrangement is to better for present the contents and make more clear for readers in their logical thinking.

Graphically' results are presented clearly but the English style of the text is poor and it  was hard to  interpret some sentences that have to be rephrased. Technical terms are often inappropriate. 

-I am so sorry to make you feel hard to interpret some sentences. I have carefully reviewed the article and corrected language mistakes again and again, thank you very much.

There is a funfamental methodological aspect that nust be addressed. It is not clear what is the Control in the experiment; which kind of soil and in which sense it can be regarded as a Control?

-This part have been rewrote to make “CK” more clearly. Thank you very much for your reminding.

I suggested to abbreviate the Title 

-The title has been abbreviated.

For more detailed text editings, comments and suggestions see notes in the text (Attached PDF file)

-The suggestions for revisions were revised one by one in the manuscript, and highlighted using blue fonts. There are still some revisions, I need to

Detailed Comments

  1. The original article  has to be split into two distinct articles :
  • Characterizations of rhizosphere microbioma of rapeseed plants

 Selection of candidate Plant Groth Promoting Rhizobacteria from the rhizosphere microbioma of rapeseed 

-Thank you for providing us an opportunity to revise the manuscript.

Characterization the diversity of rhizosphere microorganisms in different growth stages of rapeseed is an very way for us to understand the rhizosphere microenvironment of rapeseed, to further obtain soil microbial resources, and to develop useful microbial strains for practical applications.

Putting these two results together in one article is not contradictory.

After understanding the background of microbial diversity in soil, it is more convincing to develop growth promoting microorganisms

This research approach makes theoretical research more closely integrated with practical research.

  •  
  1. In the first of the two putative  articles the Authors have to make  clear what they considered as a Control . They  compared the rhizosphere microbiomaa of rapseeed at two different  phenological stages. What was the control and what was its scope?         
    --This part have been rewrote to make “CK” more clearly. Thank you very much for your reminding.

  2. In preparing the manuscripts of these two putative articles I suggest the Authors to ask a colleague of mother tonguefor help. I tried to suggest some changes  (see notes in the text, attached file available also for the Authors).

-I have carefully reviewed the article and corrected language mistakes again and again with the help of an expert, thank you very much. The suggestions for revisions were revised one by one in the manuscript, and highlighted using blue fonts.

Comments on the Quality of English Language

English language is quite poor . It was very hard to interpret what the Authors wanted to say. Some sentences have to be rephrased

-I am feel sorry to make you feel hard to interpret what the Authors wanted to say. I have carefully reviewed the article and corrected language mistakes again and again with the help of an expert, thank you very much.

Reviewer 2 Report (New Reviewer)

Comments and Suggestions for Authors

This paper focused on the area of rhizosphere microorganisms.  Authors employed a variety of methods to reveal the compositions and functions of rhizosphere microorganisms.  Soil microorganisms and their beneficial impact on plant health and mineral nutrients are a renewed interest in soil/plant research communities.  This manuscript showed a variation of soil microorganism changes at different stages of Brassica napus. This provides basic information on dynamics of rhizosphere. It also identify beneficial rhizosphere microorganisms, which might have a potential to be developed as a product for canola production. 

                  However, there are many revisions needed to be done before it can be published.  The major ones are:

1.       The “Materials and Methods is arranged after discussion, not very common in the format of journal paper.  Not so sure if this is the format required by the journal or it is a mistake by authors.  Usually, the Materials and Methods appeared right after Introduction.

2.       In the Materials and Methods, many cited methods are referred to a different reference.  This creates inconvenience for readers to comprehensively understand the paper.  Usually, a succinct description of the method is needed for the manuscript before referencing it to another one.  Also the diversity index equations are missed in the section.

3.       In the References section, each cited reference has “[crossref]”, not so sure what it means. 

Other suggestions for revisions are marked in the original manuscript.

Comments on the Quality of English Language

This paper focused on the area of rhizosphere microorganisms.  Authors employed a variety of methods to reveal the compositions and functions of rhizosphere microorganisms.  Soil microorganisms and their beneficial impact on plant health and mineral nutrients are a renewed interest in soil/plant research communities.  This manuscript showed a variation of soil microorganism changes at different stages of Brassica napus. This provides basic information on dynamics of rhizosphere. It also identify beneficial rhizosphere microorganisms, which might have a potential to be developed as a product for canola production. 

                  However, there are many revisions needed to be done before it can be published.  The major ones are:

1.       The “Materials and Methods is arranged after discussion, not very common in the format of journal paper.  Not so sure if this is the format required by the journal or it is a mistake by authors.  Usually, the Materials and Methods appeared right after Introduction.

2.       In the Materials and Methods, many cited methods are referred to a different reference.  This creates inconvenience for readers to comprehensively understand the paper.  Usually, a succinct description of the method is needed for the manuscript before referencing it to another one.  Also the diversity index equations are missed in the section.

3.       In the References section, each cited reference has “[crossref]”, not so sure what it means. 

Other suggestions for revisions are marked in the original manuscript.

Author Response

Reviewer 2

Comments and Suggestions for Authors & Comments on the Quality of English Language

This paper focused on the area of rhizosphere microorganisms.  Authors employed a variety of methods to reveal the compositions and functions of rhizosphere microorganisms.  Soil microorganisms and their beneficial impact on plant health and mineral nutrients are a renewed interest in soil/plant research communities. This manuscript showed a variation of soil microorganism changes at different stages of Brassica napus. This provides basic information on dynamics of rhizosphere. It also identify beneficial rhizosphere microorganisms, which might have a potential to be developed as a product for canola production. 

However, there are many revisions needed to be done before it can be published.  The major ones are:

-Thank you for providing us an opportunity to revise the manuscript. We have revised the manuscript as suggested. The revised parts are highlighted using blue fonts.

  1. The “Materials and Methods is arranged after discussion, not very common in the format of journal paper.  Not so sure if this is the format required by the journal or it is a mistake by authors.  Usually, the Materials and Methods appeared right after Introduction.

- The “Materials and Methods is arranged after discussion, and it’s the format required by the journal.

  1. In the Materials and Methods, many cited methods are referred to a different reference. This creates inconvenience for readers to comprehensively understand the paper.  Usually, a succinct description of the method is needed for the manuscript before referencing it to another one.  Also the diversity index equations are missed in the section.

- The contents in Materials and Methods part have been changed, and succinct descriptions of the methods have added. 

The diversity index were calculated by QIIME2 (https://qiime2.org/), then, the diversity index equations were not put in the manuscript.

  1. In the References section, each cited reference has “[crossref]”, not so sure what it means.

- The “[crossref]” in each cited reference was hyperlink address of each cited reference. I have revised the hyperlink address of each article.

Other suggestions for revisions are marked in the original manuscript.

The suggestions for revisions were revised one by one in the manuscript, and highlighted using blue fonts. There are still some revisions, I need to show here to explain clearly. Thank you very much again.

-The authors (Mengjiao Wang and Haiyan Sun) are from Shaanxi University of Technology.The official website of the school is “https://www.snut.edu.cn/”.

-“After inoculation and incubation at 28°C for 3 days,” in page 10, Line 207.

These PGPR strains in manuscript were incubated at 28°C in all experiments.

- After incubated at 28 ℃ for 18 h, the monoclonal microbes formed on medium were further purified on a petri dish using streak method.” in page 15, Line 430-431.

After inoculating the strain with an inoculum ring, a single clone was isolated by streaking on the culture medium. This is streak method, and it’s basic Operations of microbial experiments. Here, we didn’t list and references.

-“The type strains of these strains were obtained in EZ BioCloud (https://www.ezbiocloud.net), and compared with the sequences.” in page 18, Line 470-471.

EZ BioCloud (https://www.ezbiocloud.net) is website, that we could search and download the sequence of type strains. And I am so sorry to have mistake of expressing the content. The process of sequences alignment and GenBank accession numbers acquisition were rewrote in the manuscript.

-“Bray-curtis, weighted and unweighted unifrac beta diversity indexs were calculated by QIIME2 (https://qiime2.org/) and displayed with R software (V2.15.3).”  in page 18, Line 481-482.

In PCoA analysis, weighted/unweighted unifrac was used to calculate the distance between samples by using the evolutionary information of each sample sequence. In calculation of weighted unifrac, the abundance of species was considered. In calculation of unweighted unifrac, species abundances were not weighted.

-“Enterobacter spp” in page 21, Line 632, Enterobacter, a genus of bacteria, should be wrote in italics.

Reviewer 3 Report (New Reviewer)

Comments and Suggestions for Authors

In the paper "Characterizations of rhizosphere microbial diversity and Selection of plant growth-promoting rhizobacteria in flowering and fruiting stages of rapeseed" presents relevant material on the effects of bacteria and microorganisms on the rapeseed cultivation during growth and flowering stages. The aim is to produce healthier and more abundant yields by exploiting microorganisms and bacteria, while minimising negative environmental impacts.

Minor observations:

1. As many as 502 rapeseed fields were analysed, but it would be useful to mention what kind of effect could long term use of given bacteria cause.

2. I would suggest paying attention to the layout of Figures 1, 4, 5, 6, 7, 8, 9. (font size)

Author Response

Reviewer 3

Comments and Suggestions for Authors

In the paper "Characterizations of rhizosphere microbial diversity and Selection of plant growth-promoting rhizobacteria in flowering and fruiting stages of rapeseed" presents relevant material on the effects of bacteria and microorganisms on the rapeseed cultivation during growth and flowering stages. The aim is to produce healthier and more abundant yields by exploiting microorganisms and bacteria, while minimising negative environmental impacts.

-Thank you for providing us an opportunity to revise the manuscript.

Minor observations:

  1. As many as 502 rapeseed fields were analysed, but it would be useful to mention what kind of effect could long term use of given bacteria cause.

-I am so sorry that, the PGPR strains have not been added in soil yet. The experiments about effects of PGPR on plants and soil are planed. Based on the present results of PGPR strains, the effect of given bacteria were talked in the discussion part in the manuscript.

  1. I would suggest paying attention to the layout of Figures 1, 4, 5, 6, 7, 8, 9. (font size)

-The layout of Figures 1, 4, 5, 6, 7, 8, 9 have been changed, and thank you very much for suggestion.

Round 2

Reviewer 1 Report (New Reviewer)

Comments and Suggestions for Authors

The Authors addressed allm the relevant criticisms.

I futher modified the Title as generally achroyms are not used in a title  (Plant Growth Promoting Rhizobacteria)

I added a relevant reference and changed the citation order accordingly (Please check: I did not find the last reference in the text (N° 60, former N°59)

Moreover I made minor text editings

See notes in the text (Attached PDF file)

Comments on the Quality of English Language

The English style has been substantially improved

Author Response

Reviewer 1

Comments and Suggestions for Authors

The Authors addressed all the relevant criticisms.

-Thank you for providing us an opportunity to revise the manuscript.

I further modified the Title as generally achroyms are not used in a title  (Plant Growth Promoting Rhizobacteria)

-The title has been changed as your advise, thank you very much. 

I added a relevant reference and changed the citation order accordingly (Please check: I did not find the last reference in the text (N° 60, former N°59)

-The relevant reference has been added in the manuscript, and the citation order has been changed. The last reference in the text (N° 60, former N°59) has been deleted. These two references were cited in the manuscript that was removed in the last round of revisions.

Moreover I made minor text editings

See notes in the text (Attached PDF file)

-The suggestions for revisions were revised one by one in the manuscript, the revised parts are highlighted using blue fonts.

Reviewer 2 Report (New Reviewer)

Comments and Suggestions for Authors

No comments.

Author Response

Reviewer 2

Comments and Suggestions for Authors 

No comments.

-Thank you for your approval of my manuscript.

This manuscript is a resubmission of an earlier submission. The following is a list of the peer review reports and author responses from that submission.

Round 1

Reviewer 1 Report

Comments and Suggestions for Authors

I read the manuscript in title (Characterizations of rhizosphere microbial diversity and Selection of plant growth-promoting rhizobacteria in flowering and fruiting stages of oilseed rape) its about survey of PGRP strains. So, I have some suggestions

- in Line 26, 26 re-write this sentience

- In line 65, 66 The only objective of the research was to make a survey PGRP strains. So I thing this sentence must change to this area

- In line 208, 209 How did you know there is no any method for phosphorous determination an material and methods

- There are many typos and language mistakes

- In line 400 and 416 Is it correct 

- The conclusion needs to re-write. its not clear and there are not any information about the manuscript applications and who are the benfitors. Also, it needs sentence to clear the future work in this field.

- the first sentence in conclusion not complete

Comments on the Quality of English Language

- There are many typos and language mistakes

Author Response

Responses to reviewers’ comments

Reviewer #1:

I read the manuscript in title (Characterizations of rhizosphere microbial diversity and Selection of plant growth-promoting rhizobacteria in flowering and fruiting stages of oilseed rape) its about survey of PGRP strains. So, I have some suggestions

-Thank you for providing us an opportunity to revise the manuscript. We have revised the manuscript as suggested. The revised parts are highlighted using blue fonts.

- in Line 26, 26 re-write this sentience

-The sentence was deleted because the previous result was repeated by this sentence.

- In line 65, 66 The only objective of the research was to make a survey PGRP strains. So I thing this sentence must change to this area

-The sentence was changed according to your opinion.

- In line 208, 209 How did you know there is no any method for phosphorous determination an material and methods

-First, there were some methods for phosphorous determination, and I did not mention anything about that.

The sentence was changed in order to better express the meaning.

- There are many typos and language mistakes

I have carefully reviewed the article and corrected these errors, thank you very much.

- In line 400 and 416 Is it correct 

The words in line 400 and 406 were corretced, thank you very much.

- The conclusion needs to re-write. its not clear and there are not any information about the manuscript applications and who are the benfitors. Also, it needs sentence to clear the future work in this field.

The conclusion was rewritten. And thank you for your advise.

- the first sentence in conclusion not complete

The conclusion was rewritten.

Reviewer 2 Report

Comments and Suggestions for Authors

This manuscript presents interesting information. The data is mostly explained, but no results were presented showing the differences found between different plant stages. See the attached file for line by line comments.

Comments on the Quality of English Language

This needs extensive editing for English language. Some of the problems have bee addressed in the review.

Round 2

Reviewer 1 Report

Comments and Suggestions for Authors

The manuscript improved as my suggestions and authors made it all

Author Response

Responses to reviewers’ comments

Reviewer #1:

Comments and Suggestions for Authors

The manuscript improved as my suggestions and authors made it all

-Thank you very much for your approval of the article and its revisions.

Reviewer 2 Report

Comments and Suggestions for Authors

The changes improved the manuscript, but additional changes are needed. See line by line suggestions below.

Line 25 - write out Thirty four

Line 27 - change to '...stages of oilseed rape fields, respectively.'

Line 66 - change  microbial to microbiota

Line 125 - change to 'Two other genera, Fusarium and Cystofilobasidium, had a relative...'

Line 127 - write out two

Line 133 - change to '...samples indicated by the...'

Line 174- write out Twenty two

All Figure 6 citations -Figure must be capitalized

Line 267 - Please include a few sentences explaining how the microbial communities were different between flowering and fruiting stages for each soil type. Were the numbers of specific types or genera different between flowering and fruiting soils? Consider including the information into a table.

Line 274 - change flowing to flowering

Line 436 - write out Sixty five

Line 442 - change 'found' to 'determined'; change spelling of oilseed

Comments on the Quality of English Language

English has improved, but additional changes are necessary.

Author Response

Responses to reviewers’ comments

Reviewer #2:

Comments and Suggestions for Authors

The changes improved the manuscript, but additional changes are needed. See line by line suggestions below.

- Thank you for your advise for revising the manuscript. We have revised the manuscript as suggested. The revised parts are highlighted using blue fonts. 

Line 25 - write out Thirty four

- “Thirty four” has been written out.

Line 27 - change to '...stages of oilseed rape fields, respectively.'

- The sentence has been changed.

Line 66 - change microbial to microbiota

- The “microbial” has been changed to “microbiota”.

Line 125 - change to 'Two other genera, Fusarium and Cystofilobasidium, had a relative...'

- The sentence has been changed.

Line 127 - write out two

- “Two has been written out.

Line 133 - change to '...samples indicated by the...'

- The sentence has been changed.

Line 174- write out Twenty two

- “Twenty two has been written out.

All Figure 6 citations -Figure must be capitalized

- All Figure 6 citations were changed. And thank you very much.

Line 267 - Please include a few sentences explaining how the microbial communities were different between flowering and fruiting stages for each soil type. Were the numbers of specific types or genera different between flowering and fruiting soils? Consider including the information into a table.

- The sentences explaining how the microbial communities were different between flowering and fruiting stages for each soil type were added in the manuscript.  

The relative abundance of microbial community in flowering and fruiting stages of oilseed rape were significantly different from the control group. The relative abundance of microbial community indicated by mean and standard deviation were shown in Appendix Table 1. The differences between groups were analyzed by t-test. 

Line 274 - change flowing to flowering

- “ flowing” has been changed to “flowering”.

Line 436 - write out Sixty five

- “Sixty five” has been written out.

Line 442 - change 'found' to 'determined'; change spelling of oilseed

- 'found' has been changed to 'determined', and the spelling of oilseed has been changed. Thank you very much.

Round 3

Reviewer 2 Report

Comments and Suggestions for Authors

All minor editing improved the paper. Only one change needed. In Appendix 1 Table, please separate out flowering soil organisms from fruiting soil organisms and from control. Will also need to define control in the legend.

Author Response

Responses to reviewers’ comments

Reviewer #2:

Comments and Suggestions for Authors

All minor editing improved the paper. Only one change needed. In Appendix 1 Table, please separate out flowering soil organisms from fruiting soil organisms and from control. Will also need to define control in the legend.

- Thank you for your advise for revising the manuscript. Appendix 1 Table has been revised in the manuscript as suggested. The revised parts are highlighted using blue fonts. 
